# Modelling and RBF Control of Low-Limb Swinging Dynamics of a Human–Exoskeleton System

**Xinyu Peng** †, **Shujun Zhang** † , **Mengling Cai and Yao Yan** *

School of Aeronautics and Astronautics, University of Electronic Science and Technology of China, Chengdu 611731, China
* Correspondence: y.yan@uestc.edu.cn
† These authors contributed equally to this work.

**Abstract:** With the increase in the elderly population in China and the growing number of individuals who are unable to walk normally, research on lower limb exoskeletons is becoming increasingly important. This study proposes a complete dynamic model parameter identification scheme for the human–machine coupling model of lower limb exoskeletons. Firstly, based on the coupling model, the excitation trajectory is optimized, data collection experiments are conducted, and the dynamic parameter vector of the system is identified using the least squares method. Secondly, this lays the foundation for designing adaptive control based on RBF neural network approximation. Thirdly, the Lyapunov function is used to prove that the RBF neural network adaptive controller can achieve stable tracking of the lower limb exoskeleton. Finally, simulation analysis reveals that increasing the gains of the RBF controllers effectively reduces tracking errors. Furthermore, the tracking errors and control torques show that adaptive control based on the RBF neural network approximation works well.

**Keywords:** lower limb exoskeleton; parameter identification; the least squares method; RBF neural network adaptive controller





## 1. Introduction

According to some references, by 2030, 18.2% of Chinese people will be over 65 years old [1]. Additionally, the number of people with physical disabilities will reach 24.12 million, accounting for 29.07% of the total number of disabled individuals, among which there are approximately 1.58 million people with lower limb paralysis [2]. Currently, China is becoming an aging society, and with the growth of the elderly population and the occurrence of various accidents, the number of people who have difficulty in walking is increasing year by year. However, due to various reasons, only a small portion of the population can receive timely and effective rehabilitation treatment [3]. This can cause significant potential harm to the lives of the patients and their families [4]. At this point, the emergence of lower limb exoskeletons brings significant help to patients and their families. Lower limb exoskeletons can assist patients in effective rehabilitation training, improve their lower limb motor function, and alleviate the burden on the patients' caregivers. The emergence of lower limb exoskeletons brings new hope to patients and improves their quality of life [5]. Generally speaking, lower limb exoskeletons have two different objectives: assisting patients in rehabilitation training, and assisting in human working activities [6]. Robotic technologies that assist physical health and provide support for the elderly are rapidly developing. In the past three decades, lower limb exoskeleton robotics and assistive technologies have been a focus of attention in the industry [7]. As a typical application of human–machine interaction devices, the perception and control of lower limb exoskeleton robots will significantly affect the actual wearing effects of users in lower limb assistance or augmentation [8]. Therefore, in order to achieve the swinging control of

the lower limb exoskeleton, it is essential to design an appropriate swing controller during the development stage [9].

Currently, an increasing number of researchers aim to enhance the development and improvement of lower limb exoskeletons through control, thereby improving their performance [10]. With the rise of model-based control techniques, knowledge of the precise dynamic mathematical model of lower limb exoskeletons is required. However, in practical systems, various uncertainties exist that have a significant impact on the model-based controllers, thereby severely affecting their performance. Therefore, designing a reasonable control algorithm to address the uncertainties in lower limb exoskeleton systems has become a research hotspot [11]. Currently, optimization control strategies for lower limb exoskeletons need to meet the criteria of safety, stability, effective rehabilitation, assistance, and efficient control simultaneously [12]. Furthermore, for lower limb exoskeletons, controllers need to demonstrate good tracking performance and stability while minimizing tracking errors [13]. Therefore, appropriate control algorithms are required to ensure the stability and robustness of lower limb exoskeletons during motion [14]. Currently, the main control strategies for lower limb exoskeletons include position tracking control, force impedance control, biological signal control, and more. Among them, position tracking control serves as the foundation for other control methods [15]. Additionally, the control modes of exoskeletons primarily involve active control and passive control. In active control, the exoskeleton provides necessary assistance and closely follows human motion. In passive control, the human body acts as the load for the exoskeleton, and the exoskeleton is entirely driven by external forces. The key aspect of passive control lies in studying the dynamic model of the exoskeleton [16]. From this perspective, the development of control systems is one of the most critical parts of exoskeleton systems. RBF neural networks are chosen due to their excellent generalization ability, simple structure, and the ability to approximate any nonlinear function with arbitrary precision. They can be used to estimate the unknown parts of the hyper-local model [17]. Reference [18] considers the exoskeleton as a nonlinear motion system and applies RBF neural networks for real-time identification of the system. The current hip and ankle joint angles of the exoskeleton, along with the previous knee joint angle, are defined as the network inputs at the current time. A general error function is defined, which includes tracking errors of the exoskeleton and interaction torques between the pilot and the exoskeleton to ensure convergence. Subsequently, a mathematical model of the human–machine system is established. It can be seen that the RBF neural network controller performs well in controlling the lower limb exoskeleton robot through simulation experiments.

Furthermore, through extensive literature review, it is evident that determining the inertial parameters of the robot is necessary to develop advanced control algorithms. The design of nonlinear robot controllers typically relies on the robot model, and their performance directly depends on accurate inertial parameters. Due to the complexity of certain robot structures and the nonlinear nature of loads, determining dynamic parameters can be challenging, and parameter identification is the only effective method to obtain precise inertial parameters [19]. As model-based control is crucial, the identification of inertial parameters has gained extensive attention from researchers [20]. Identifying the inertial parameters first requires establishing a mathematical model and linearizing it to obtain a coefficient matrix for parameter identification [21]. In the offline dynamic identification stage the coefficient matrix is utilized, and the parameters are calculated using the least squares method [22].

It can be seen that the control of lower limb exoskeletons is crucial in their research. Model-based control methods have been used to achieve a desirable control performance. This study focuses on lower limb exoskeleton robots. Considering the individual differences among people, this work firstly determines the inertial parameters of the human–machine coupling model through experiments to establish a foundation for control simulations, rather than using average anatomical values as the model's inertial parameters. In the control simulation stage, assuming the exoskeleton operates in passive control mode, an

adaptive controller based on RBF neural network approximation is designed to track the desired trajectory. Simulations are conducted using MATLAB to indicate the control effects of the RBF neural network controller.

## 2. Materials and Methods

### 2.1. Coupled Dynamics Model of 2-DOF Lower Extremity Exoskeleton and Human Body

In this paper, a Lagrange modelling approach is utilized to research the dynamics of a lower limb exoskeleton robot and the human body. The Lagrange method firstly finds the total kinetic energy $E_k$ and the potential energy $E_p$ of the model, and then puts $E_k$ and $E_p$ into the Lagrange function to get $L_a = E_k - E_p$. Then, the Lagrange function equation can be derived from its partial derivation to find out the magnitude of the actuating torque needed to rotate the corresponding joints in the model.

Figure 1 shows the schematic diagram of a 2-DOF lower extremity exoskeleton coupled with the human body. The corresponding symbols for the physical parameters in the figure are shown in Table 1. We treat the human body and lower limb exoskeleton as a whole in this section.

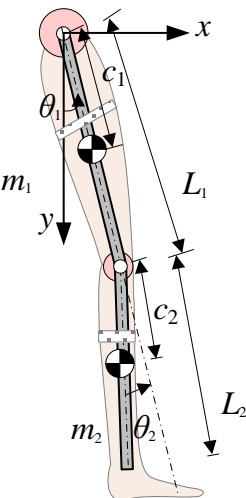

**Figure 1.** Model of Lower Extremity Exoskeleton and Human Body.

**Table 1.** Symbolic representation of physical parameters for the coupled model.

| Meaning of Parameters | Notation |
|---|---|
| Hip joint angle/Knee joint angle | $\theta_1/\theta_2$ |
| Thigh length/Shank length | $L_1/L_2$ |
| Thigh mass/Shank mass | $m_1/m_2$ |
| Thigh center of mass/Shank center of mass | $c_1/c_2$ |
| Thigh moment of inertia/Shank moment of inertia | $I_1/I_2$ |

According to the kinetic energy formula, the translational kinetic energy and the rotational kinetic energy of the center of mass, respectively, make up the kinetic energy of the thigh $E_{k_1}$ and the shank $E_{k_2}$:

$$E_{k_1} = \frac{1}{2}m_1 c_1^2 \dot{\theta}_1^2 + \frac{1}{2}I_1 \dot{\theta}_1^2 \tag{1}$$

$$E_{k_2} = \frac{1}{2}m_2(l_1^2\dot{\theta}_1^2 + c_2^2\dot{\theta}_1^2 + c_2^2\dot{\theta}_2^2 + 2L_1c_2\dot{\theta}_1^2\cos\theta_2 \\ + 2L_1c_2\dot{\theta}_1\dot{\theta}_2\cos\theta_2 + 2c_2^2\dot{\theta}_1\dot{\theta}_2) + \frac{1}{2}I_2(\dot{\theta}_1 + \dot{\theta}_2)^2 \tag{2}$$

From the potential energy equation, the potential energy of the thigh $E_{p_1}$ and the shank $E_{p_2}$ can be obtained:

$$E_{p_1} = -c_1 m_1 g \cos\theta_1 \tag{3}$$

$$E_{p_2} = -m_2 g [L_1 \cos\theta_1 + c_2 \cos(\theta_1 + \theta_2)] \tag{4}$$

where gravitational acceleration $g = 9.8 \ m/s^2$.

Then the Lagrange function $L_a$ can be written as:

$$L_a\left(\theta_1, \theta_2, \dot{\theta}_1, \dot{\theta}_2\right) = E_{k_1} + E_{k_2} - E_{p_1} - E_{p_2} \tag{5}$$

Lagrange dynamic method is used to solve the system's equations, and the Lagrange equation is:

$$T = \frac{\partial}{\partial t} \frac{\partial L_a}{\partial \dot{\theta}} - \frac{\partial L_a}{\partial \theta} \tag{6}$$

where $\theta = [\theta_1, \theta_2]^T \in \mathbb{R}^2$ represents the dual angle vector representing the coupled model, $T$ represents the joint generalized vector.

According to the above derivation, the dynamic model of the coupled human lower extremity exoskeleton can be represented as:

$$M(\theta)\ddot{\theta} + C(\theta, \dot{\theta})\dot{\theta} + G(\theta) + \tau_{fic} = \tau \tag{7}$$

In Equation (7), $M(\theta)$, $C(\theta, \dot{\theta})$ and $G(\theta)$ represent the Inertia matrix, Coriolis matrix and Gravity matrix, $\tau_{fic} \in \mathbb{R}^2$ represent the dual joint friction torque, $\tau \in \mathbb{R}^2$ represent the dual joint control torque.

Convert the dynamic model into a linear form:

$$M(\theta)\ddot{\theta} + C(\theta, \dot{\theta})\dot{\theta} + G(\theta) + \tau_{fic} = A(\theta, \dot{\theta}, \ddot{\theta})X = \tau \tag{8}$$

In Equation (8), $X \in \mathbb{R}^8$ represents the parameter vector to be identified, $A(\theta, \dot{\theta}, \ddot{\theta}) \in \mathbb{R}^{2 \times 8}$ represents the regression matrix composed of the angle, angular velocity, and angular acceleration of the hip and knee joints. The specific forms of the parameter vector to be identified and the regression matrix are provided in Appendix A.

Assuming that there are $N$ sets of sampling points in the experiment, the sampling regression matrix $\overline{A} \in \mathbb{R}^{2N \times 8}$ and the sampling torque vector $\overline{\tau} \in \mathbb{R}^{2N \times 1}$ can be represented as:

$$\overline{A} = \begin{bmatrix} A^{(1)} \\ A^{(2)} \\ \vdots \\ A^{(N)} \end{bmatrix}, \ \overline{\tau} = \begin{bmatrix} \tau^{(1)} \\ \tau^{(2)} \\ \vdots \\ \tau^{(N)} \end{bmatrix} \tag{9}$$

The parameter vector to be identified can be obtained using the least square method:

$$\hat{X} = (\overline{A}^T \overline{A})^{-1} \overline{A}^T \overline{\tau} \tag{10}$$

Due to the presence of signification noise in the sampled data, designing a suitable excitation trajectory can help suppress the noise and thereby improve the accuracy of the parameter vector to be identified.

The excitation trajectory can be represented using a second-order Fourier series:

$$\begin{aligned} \theta_d &= \theta_0 + \sum_{k=0}^{\overline{n}} (a_{i,k} \sin(k\omega t) + b_{i,k} \cos(k\omega t)) \\ \dot{\theta}_d &= \sum_{k=0}^{\overline{n}} (a_{i,k} k\omega \cos(k\omega t) - b_{i,k} k\omega \sin(k\omega t)) \\ \ddot{\theta}_d &= \sum_{k=0}^{\overline{n}} \left( -a_{i,k}(k\omega)^2 \sin(k\omega t) - b_{i,k}(k\omega)^2 \cos(k\omega t) \right) \end{aligned} \tag{11}$$

In Equation (11), $i = 1, 2$, $k = 1, \cdots, \overline{n}$, $t \in [0, T]$ and $T$ represents the duration of the data collection experiment for design, $\omega$ represents the set base frequency, $k$ represents the set sampling parameter, $\theta_{i,0}$ represents the initial bias angle of the exoskeleton hip and knee joints, and $a_{i,k}$ and $b_{i,k}$ represent the parameters to be optimized. Considering the joint angle limitation range during human movement [23] as well as our hardware device restriction, the constraint intervals for the excitation trajectory are shown in Table 2.

**Table 2.** The constraint intervals for the excitation trajectory.

| Constraint Terms | Constraint Intervals |
|---|---|
| Hip joint angle/ Knee joint angle (rad) | $[0.0872, 1.2217]/[-1.6232, -0.2269]$ |
| Hip/Knee angular velocities (rad/s) | $[-1.6449, 1.6449]$ |
| Hip/Knee angular accelerations (rad/s$^2$) | $[-5.1677, 5.1677]$ |

In MATLAB, the Equation (11) is sampled to obtain a sampled regression matrix. By using the particle swarm algorithm to optimize the obtained matrix, the parameters $a_{i,k}$ and $b_{i,k}$ to be optimized can be obtained. The condition number of the sampling regression matrix is selected as the fitness function ($Cond(\overline{A})$) for the optimization process. Meanwhile, the set excitation trajectory should follow the constraint conditions shown in Table 2. Based on the final set, the excitation trajectory is as follows:

$$Fit(\overline{A}) = \begin{cases} Cond(\overline{A}) & \text{if} \begin{cases} 0.0872 < \theta_1 < 1.2217 \\ 1.6232 < \theta_2 < -0.2269 \\ 1.6449 < \dot{\theta}, \ddot{\theta} < 1.6449 \end{cases} \\ Inf & \text{Otherwise} \end{cases} \tag{12}$$

### 2.2. Adaptive Control of Exoskeleton Based on RBF Neural Network Approximation

The most classic control algorithms for exoskeletons include PID control, neural network control, fuzzy control, iterative learning control, and robot inversion control [24]. The PD controller is the most widely used control algorithm, especially in a nonlinear control system. The PD controller's leading characteristics can be utilized to enhance the dynamic performance and robustness of the control system [25]. RBF neural networks, due to their good performance and simple structure, can avoid unnecessary and complex calculations that are commonly used in a nonlinear system [26]. Compared to the PD controller, the RBF neural network can eliminate the error caused by disturbance and accelerate convergence speed [27].

In practical application engineering, there may be unknown external disturbance acting on the torque $\tau_d$ applied to the exoskeleton [28]. When an unknown external disturbance is added to system, the mathematical expression for the model of the limb exoskeleton is:

$$M\ddot{\theta} + C\dot{\theta} + G = \tau - \tau_d - \tau_{fric} \tag{13}$$

The $M$, $C$, $G$ matrices represent the inertia matrix, Coriolis acceleration, and gravity matric of the identified exoskeleton and human body system.

The input of the system $\theta_d(t)$ is the desired angle vector for the exoskeleton's hip and knee joints, and the output $\theta(t)$ is the actual tracked angle vector for the exoskeleton's hip and knee joints. Therefore, the error of the system is given by:

$$e(t) = \theta_d(t) - \theta(t) \tag{14}$$

In order to achieve a better control performance, it is necessary to compensate for the unknown disturbances. Because the RBF neural network has a good error compensate capability, it is used to compensate for the unknown external disturbances in the exoskeleton. The error function is defined as:

$$q = \dot{e} + \Lambda e \tag{15}$$

In Equation (15) $\Lambda = \Lambda^T > 0$, which represents the magnification. By combining Equations (14) and (15), we obtain:

$$\dot{\theta} = -q + \dot{\theta}_d + \Lambda e \tag{16}$$

Taking the derivative of Equation (16) with regards to time and left-multiplying by the matrix $M$, we obtain:

$$M\dot{q} = M(\ddot{\theta}_d - \ddot{\theta} + \Lambda \dot{e}) = M(\ddot{\theta}_d + \Lambda \dot{e}) - M\ddot{\theta} \tag{17}$$

Substituting Equations (13) to (17) into the previous equation, we obtain:

$$M\dot{q} = -Cq - \tau + f + \tau_d \tag{18}$$

The expression for $f$ in Equation (18) is:

$$f = M(\ddot{\theta}_d + \Lambda e) + C(\theta_d + \Lambda e) + G + \tau_{fric} \tag{19}$$

The unknown $f$ represents the model uncertainty in practical engineering applications, and it needs to be approximated. Equation (19) indicates that this uncertainty is a nonlinear function, which can be approximated using an RBF neural network. The chosen RBF neural network is a three-layer feedforward network with a single hidden layer, which is more straightforward and computationally straightforward than higher level neural networks, while still satisfying our model's control requirements. The transformation from the input layer to the hidden layer is nonlinear. The input signals of the network are denoted as $x = [e^{\mathrm{T}} \quad \dot{e}^{\mathrm{T}} \quad q_d^{\mathrm{T}} \quad \dot{q}_d^{\mathrm{T}} \quad \ddot{q}_d^{\mathrm{T}}]$. The transformation function in the hidden layer is the radial basis function, represented as $h_j$, $j = 1, 2, \ldots,$ m. The radial basis function is used in the Gaussian kernel function, and its expression is:

$$h_j(x) = \exp[-\frac{\|x - c_j\|^2}{b_j^2}] \tag{20}$$

In Equation (20), $c_j$ represents the center of the $j$-th radial basis function, and $b_j$ represents the width of the $j$-th radial basis function.

The transformation from the hidden layer to the output layer in the RBF neural network is a linear transformation. The output expression is as follows:

$$\hat{f} = \hat{W}^T H(x) \tag{21}$$

Let

$$\widetilde{W} = W - \hat{W} \tag{22}$$

In Equation (22), $\|W\|_F \leq W_{max}$ and $\hat{W}^T$ represent the weight matrix of the RBF neural network, while $H(x)$ represents the output matrix of the radial basis functions.

Since the role of the neural network is to compensate for the model uncertainty, the control law of the designed RBF neural network controller is as follows:

$$\tau = \hat{W}^T H(x) = K_v q - v \tag{23}$$

In Equation (23), $K_v$ represents the amplification coefficient of the error, while $v$ represents the robust term used to overcome approximation errors in the neural network.

### 2.3. Adaptive Analysis of Neural Network Stablility and Converagence

By substituting the control law (23) into Equation (18) and simplifying, we obtain:

$$M\dot{q} = -(K_v + C)q + \xi_1 \tag{24}$$

where $\xi_1 = \widetilde{W}^T H(x) + (\varepsilon + \tau_d) + v$.

Designing the robust term $v$ as follows:

$$v = -(\varepsilon_N + \tau_d)\mathrm{sgn}(q) \tag{25}$$

In Equation (25), $\varepsilon_N$ and $b_d$ represent the upper bounds of $\|\varepsilon\|$ and $\tau_d$.

The expression for the Lyapunov function is defined as:

$$L = \frac{1}{2}q^T M q + \frac{1}{2}tr(\widetilde{W}^T F^{-1}\widetilde{W}) \tag{26}$$

In Equation (26), $F$ is a positive definite matrix, and taking the derivative of Equation (26), we obtain:

$$\dot{L} = q^T M \dot{q} + \frac{1}{2}q^T \dot{M} q + tr(\widetilde{W}^T F^{-1}\dot{\widetilde{W}}) \tag{27}$$

By substituting Equation (22) into the previous equation, we obtain:

$$\dot{L} = -q^T K_v q + \frac{1}{2}q^T (\dot{M} - 2C)q + tr\widetilde{W}^T (F^{-1}\dot{\widetilde{W}} + Hq^T) + q^T(\varepsilon + \tau_d + v) \tag{28}$$

Taking into account the characteristics of the exoskeleton, we choose the adaptive law of the neural network as:

$$\dot{\widetilde{W}} = FHq^T \tag{29}$$

Therefore:

$$\dot{L} = -q^T K_v q + q^T(\varepsilon + \tau_d + v) \tag{30}$$

Due to:

$$q^T(\varepsilon + \tau d + v) = q^T(\varepsilon + \tau_d) + q^T v = q^T(\varepsilon + \tau_d) - \|q\|(\varepsilon_N + \tau_d) \leq 0 \tag{31}$$

Based on the above analysis, we can conclude that: $\dot{L} \leq 0$. When $\dot{L} \equiv 0$, $q \equiv 0$, according to LaSalle's invariance principle, the closed-loop system is asymptotically stable, when $t \rightarrow \infty$, $q \rightarrow 0$ and the tracking error $e$ corresponds to zero.

## 3. Results

### 3.1. Identification of Human–Machine Coupling Parameters

The Particle Swarm Optimization (PSO) algorithm is used to optimize the excitation trajectory, with the chosen parameters of $\omega = 0.2\pi$, $\overline{n} = 2$ and $\theta_0 = [0.611, -0.925]^T$. To sample Equation (11) in MATLAB, the sampling regression matrix $\overline{A}$ is obtained. The condition number of the sampling regression matrix is calculated. By utilizing the PSO algorithm, the parameters $a_{i,k}$ and $b_{i,k}$ can be optimized to obtain the desired results. The optimization results based on the PSO are as follows: $a_{1,1} = 0.0229$, $b_{1,1} = -0.0006$, $a_{1,2} = -24.0422$, $b_{1,2} = 17.9290$, $a_{2,1} = 13.0688$, $b_{2,1} = 22.6565$, $a_{2,2} = -3.0457$, $b_{2,2} = -15.1847$, and the fitness at this moment is 11.6060.

In the data sampling experiment, a model-free PD controller and the UEXO-I Lower Limb Exoskeleton Prototype is used. The optimized excitation trajectory is input into the computer to enable the exoskeleton to swing according to the optimized trajectory. In the first step of the experiment, the exoskeleton swings alone under the control of the PD controller, without any participation of experimental personnel. This step aims to separate the friction parameters. As seen in Figure 2, two servo motor actuators (GDM1-100N2/120N2) and two motor drivers (Elmo-G-SOLHOR15/100EE) drove the exoskeleton's thigh and shank. Two absolute encoders (INC-4-150 and INC-3-125) were used to measure the rotational angles of the robot's hip and knee, and four 3D force sensors (JNSH-2-10kg-BSQ-12) were used to detect the coupling forces.

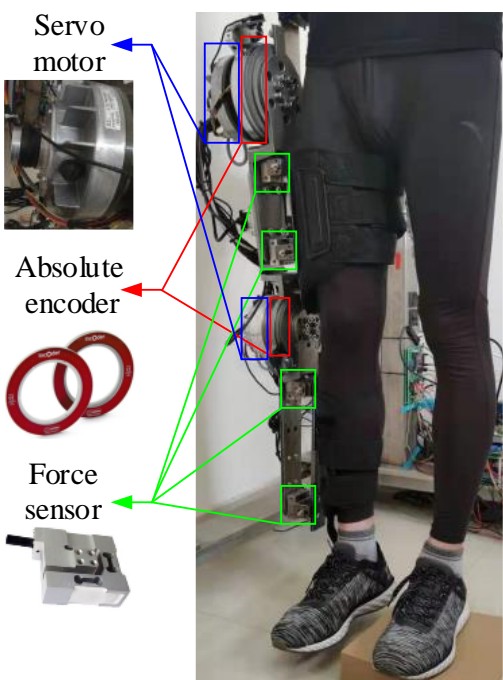

**Figure 2.** The exoskeleton used for experiment.

In the second step, the experimental subject, an 88 kg/1.73 m healthy adult, is driven by the exoskeleton without actively exerting force. The subject was standing during the trial, enabling the entire leg to swing and the knee to be in a natural posture. This step is used to determine the inertia parameters of the human body and exoskeleton. Ethical statements: The subject gave informed consent for inclusion before their participation in the study. The study was conducted in accordance with the Declaration of Helsinki, and protocol was approved by the Ethics Committee of the University of Electronic Science and Technology of China (106142023021725090). The parameters obtained from the human–machine identification experiment are shown in Table 3.

**Table 3.** Identification parameters of human body and exoskeleton system.

| Parameter | Value | Parameter | Value |
| --- | --- | --- | --- |
| $X_1$ | 10.3016 | Hip joint's coefficient of static friction ($Ks_1$) | 2.6596 |
| $X_2$ | 3.0157 | Hip joint's coefficient of viscous friction ($K_{m1}$) | 4.14895 |
| $X_3$ | 2.2664 | Knee joint's coefficient of static friction ($Ks_2$) | 4.7197 |
| $X_4$ | 5.5354 | Knee joint's coefficient of viscous friction ($K_{m2}$) | 7.3602 |

The root mean square error between the actual sampled torque of the hip joint and the estimated torque obtained by substituting the identified parameters into Equation (2) is 6.9395. Similarly, the root mean square for the knee joint is 10.2776.

### 3.2. Simulation Experiment of RBF Control

The simulation of RBF control was conducted using Simulink. The required parameters for the neural network were obtained from reference [28]. The desired trajectory settings were obtained from reference [29]. Furthermore, considering that commercially available motors generally provide a maximum torque of exceeding 200 (N·m), the maximum torque output of the controller was set to 200 (N·m).

In the control of the RBF neural network, the parameter $K_v$ amplifies the error function, and the selection of an appropriate $K_v$ directly affects the control effect. The influence of $K_v$ on the maximum absolute error of the system after stabilization is shown in Figure 3.

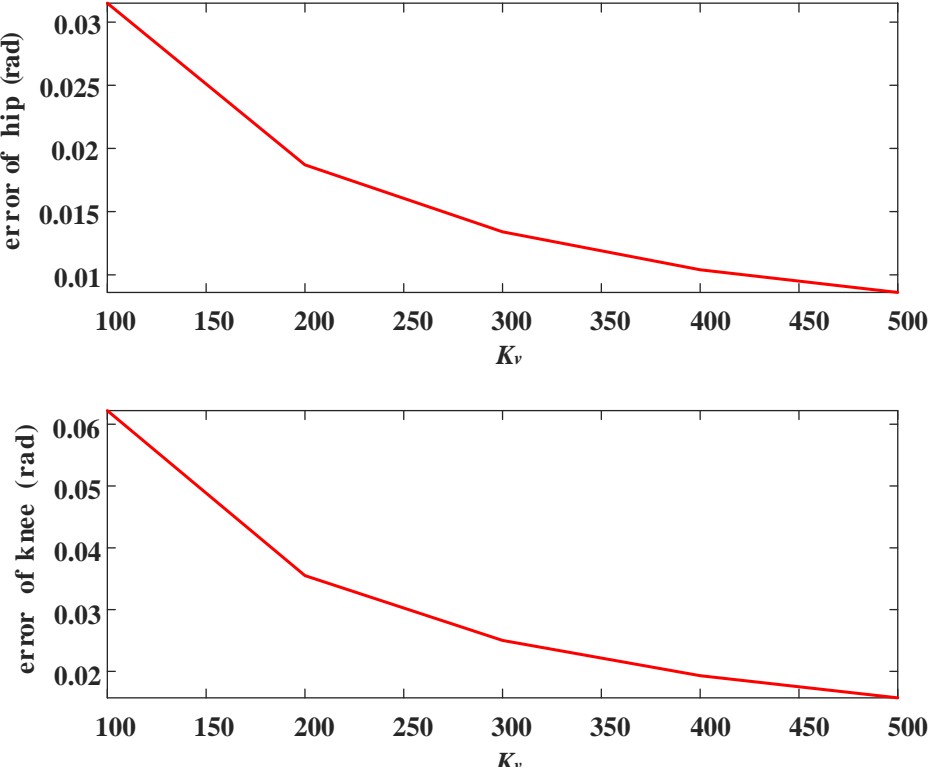

**Figure 3.** Relationship between $K_v$ and the maximum absolute value.

In Figure 3, it can be observed that increasing $K_v$ initially leads to a rapid decrease in the maximum absolute error of the hip and knee joints. It can be observed that the error decreases rapidly when $K_v$ is between 100 and 200. However, once $K_v$ increases to 300, the rate of error reduction becomes very slow. Increasing $K_v$ further from 300 to 500 does not result in an error reduction of less than 0.01 (rad). However, once $K_v$ exceeds 300, the error reduction becomes very slow. Additionally, in the simulation experiment it was discovered that when $K_v$ is too large the system's noise is amplified, resulting in poor system performance. When selecting $K_v$, it is important to ensure that the system's error does not decrease sharply with increasing $K_v$ and that the system is not affected by noise. Therefore, $K_v = 300$ was chosen.

Suitable gains were selected for RBF control and simulation verification was concluded in MATLAB. The comparison between the desired trajectory and the actual tracking trajectory of the system under the RBF neural network control is shown in Figure 4.

In Figure 4, it can be observed that the RBF neural network control can track the desired trajectory of the system. From the graph, it is evident that the RBF control can quickly track the desired trajectory in less than 1 s at the initial stage. The errors between the actual trajectory and the desired trajectory under the RBF neural network control are shown in Figure 5.

In Figure 5, it can be observed that the control performance of the hip joint is superior to that of the knee joint. In the control of the hip joint, the RBF neural network control shows almost no error after reaching a steady state, with the error approaching zero. This is because in the RBF neural network control, the network adaptive law is updated at each sampling instance, enabling dynamic adjustment of the system and effectively compensating for uncertainties in the exoskeleton model. The unknown uncertainties—including noise—that can be approximated by the RBF network come from the actual environment [28]. Therefore, the RBF neural network control demonstrates superior performance in tracking the desired trajectory.

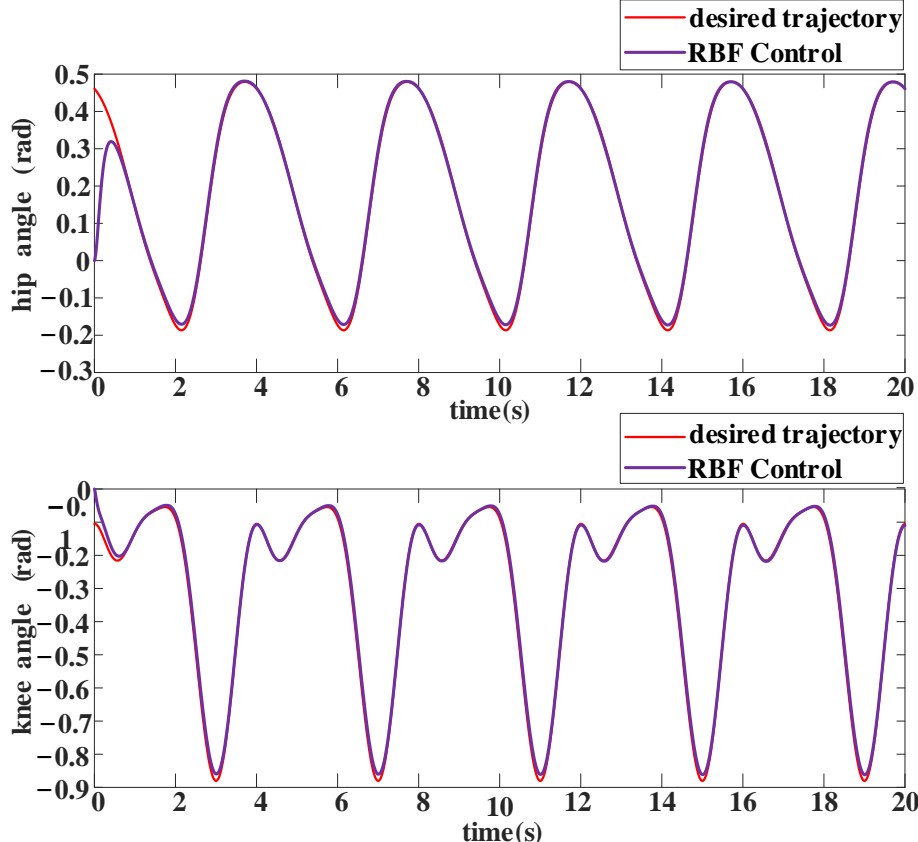

**Figure 4.** Comparison of trajectories between the desired trajectory and RBF neural network.

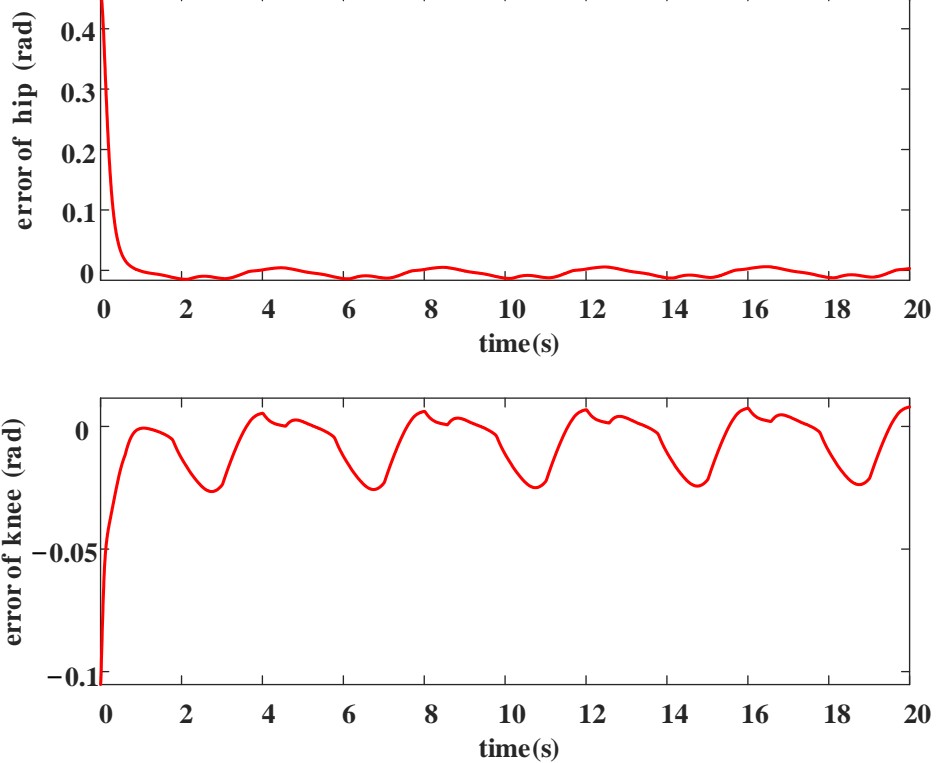

**Figure 5.** Error in RBF neural network control.

The torque required for tracking the desired trajectory in the RBF neural network control is shown in Figure 6. At the beginning stage, in order to track the desired trajectory quickly, the controller requires a large amount of torque. From the graph, it can be observed that the maximum torque in the RBF control reaches the set upper limit of 200 (N·m). If no limiting control is applied at this point, the excessively large pulses may pose certain risks to the human body and the exoskeleton.

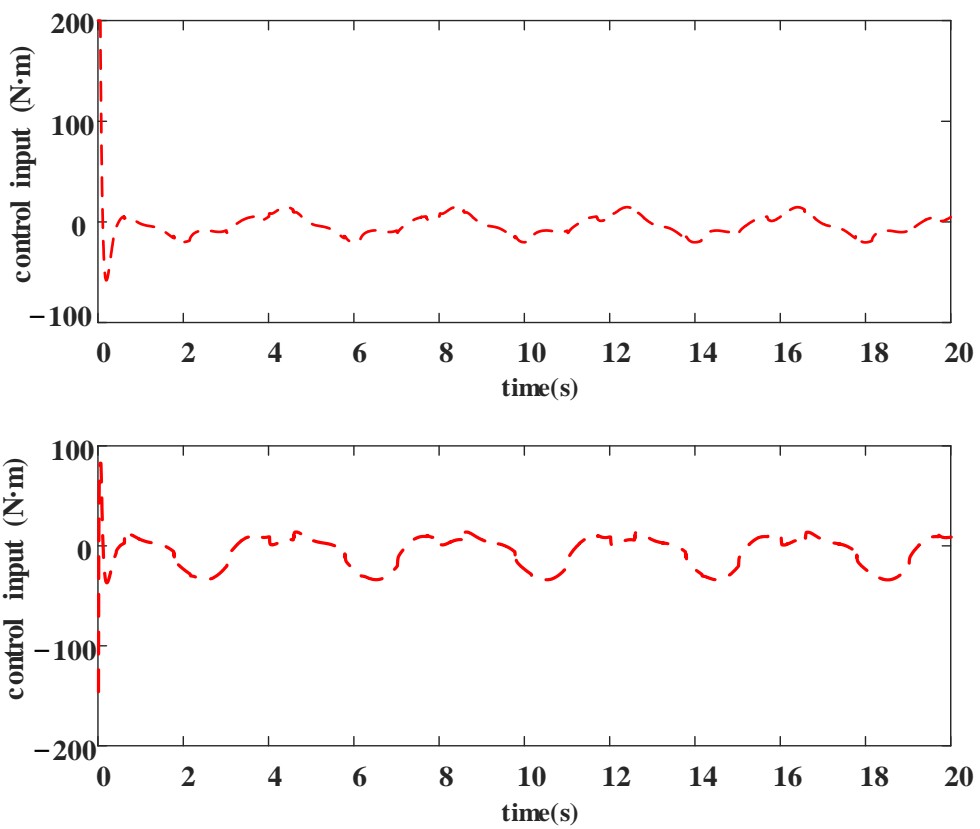

**Figure 6.** Control torque on RBF neural network control.

## 4. Discussion and Conclusions

This study focuses on the exoskeleton human–machine system. To enhance the control effectiveness of the model-based control, an identification experiment is used to obtain the inertial parameters of the human–machine system. Based on this, a simulation is made on the control effects of the RBF adaptive control. In the identification stage, firstly, the exoskeleton human–machine mathematical model is linearized and the excitation trajectory is optimized to use the particle swarm optimization algorithm. Secondly, experimental data is collected to obtained the identification parameters required for the mathematical model. Then, the inertial parameters of the human–machine model are obtained through the least square method, which helps determine the parameters for subsequent simulation experiments. Subsequently, MATLAB is used to simulate and compare the exoskeleton human–machine model. The relationship between the maximum error after stabilization and the control parameters are analyzed to select to the control gain parameters for RBF neural network control, optimizing the control for the control method. The simulation is conducted using MATLAB's S-function, and the comparative graphs of the desired and actual trajectories of the exoskeleton's hip and knee joints, error comparison, and torque comparison are obtained. Based on the dynamic compensation ability of RBF in handing system uncertainties, where control is based on the system error and its derivative, comprehensive analysis of the simulation results indicates that RBF provides desirable trajectory tracking control for the exoskeleton when an appropriate parameter is selected. The error decreases when increasing the RBF parameter $K_v$ within a specified range. However, this

reducing impact becomes smaller as the parameter values rise. From the results, we can judge that RBF achieve controlling consequences with fast convergence as well as small errors. In addition, the control performance of the hip joint is superior to that of the knee joint.

Substantial progress has been made in the research of lower limb exoskeleton robot control over the past few decades. However, whether the control strategies of lower limb exoskeletons can more effectively stimulate the functional recovery of patients remains an unresolved question. Future research should focus on structured and standardized studies aimed at identifying the relationship between control strategies and a set of core clinical outcome measures, taking into account the impact of participants' initial impairment level and training intensity [30]. In this study, we assumed that the patient does not exert any force and is completely driven by the exoskeleton, without considering the patient's intentions. In future research, we will take the patient's intentions into account and conduct in-depth investigations into the influencing factors. We plan to carry out further research on active control in the future.

**Author Contributions:** Conceptualization, X.P. and Y.Y.; methodology, X.P.; software, S.Z.; formal analysis, M.C.; writing—original draft preparation, X.P. and S.Z.; writing—review and editing, Y.Y.; visualization, Y.Y.; supervision, Y.Y.; project administration, Y.Y.; funding acquisition, Y.Y. All authors have read and agreed to the published version of the manuscript.

**Funding:** This research was funded by the National Natural Science Foundation of China (Grants No. 12072068, 11872147, 11932015, and 52175046), and the Sichuan Science and Technology Program (Grant No. 2023YFG0050).

**Data Availability Statement:** The numerical and experimental data sets generated and analyzed during the current study are available from the corresponding author on reasonable request.

**Conflicts of Interest:** The authors declare no conflict of interest.

**Appendix A**

The expression for $M(\theta)$, $C\left(\theta, \dot{\theta}\right)$, $G(\theta)$ and $\tau_{fic}$ in the dynamic model for human–machine coupling are as follows:

$$M = \begin{bmatrix} M_{11} & M_{12} \\ M_{21} & M_{22} \end{bmatrix}, C = \begin{bmatrix} C_{11} & C_{12} \\ C_{21} & C_{22} \end{bmatrix}, G = \begin{bmatrix} G_1 \\ G_2 \end{bmatrix},$$

$$M = \begin{bmatrix} X_1 + 2X_3 L_1 \cos(\theta_2) & X_2 + X_3 L_1 \cos(\theta_2) \\ X_2 + X_3 L_1 \cos(\theta_2) & X_2 \end{bmatrix}$$

$$C = \begin{bmatrix} -2X_3 L_1 \dot{\theta}_2 \sin(\theta_2) & -X_3 L_1 \dot{\theta}_2 \sin(\theta_2) \\ X_3 L_1 \dot{\theta}_1 \sin(\theta_2) & 0 \end{bmatrix}$$

$$G = \begin{bmatrix} X_3 g \sin(\theta_1 + \theta_2) + X_4 g \sin(\theta_2) \\ X_3 g \sin(\theta_1 + \theta_2) \end{bmatrix}$$

$$\tau_{fic} = \begin{bmatrix} K_{S1} \mathrm{sgn}(\dot{\theta}_1) + K_{m1} \dot{\theta}_1 \\ K_{S2} \mathrm{sgn}(\dot{\theta}_2) + K_{m2} \dot{\theta}_2 \end{bmatrix}$$

$$X_1 = m_1 c_1^2 + I_1 + m_2 L_1^2 + m_1 c_2^2 + I_2$$
$$X_2 = m_2 c_2^2 + I_2$$
$$X_3 = m_2 c_2$$
$$X_4 = m_1 c_1 + m_2 L_1$$

In the above equation, subscript 1 represents the parameters of the thigh and subscript 2 represents the parameters of the shank. By measuring the length of the exoskeleton's thigh, $L_1 = 0.38952$. In this context, g represents the gravitational constant.

For the linear form of the dynamic model, the specific expression of the regression matrix A can be represented as:

$$A = \begin{bmatrix} A_{11} & A_{12} & A_{13} & A_{14} & A_{15} & A_{16} & A_{17} & A_{18} \\ A_{21} & A_{21} & A_{22} & A_{23} & A_{24} & A_{25} & A_{26} & A_{27} \end{bmatrix}$$

$$A_{11} = \ddot{\theta}_1$$

$$A_{12} = \ddot{\theta}_2$$

$$A_{13} = L_1(2\ddot{\theta}_1\cos(\theta_2) + \ddot{\theta}_2\cos(\theta_2) - 2\dot{\theta}_1\dot{\theta}_2\sin(\theta_2) - \dot{\theta}_2^2\sin(\theta_2))$$
$$+ g\sin(\theta_1 + \theta_2)$$

$$A_{14} = g\sin\theta_1$$

$$A_{15} = \text{sgn}(\dot{\theta}_1)$$

$$A_{16} = \dot{\theta}_1$$

$$A_{17} = A_{18} = A_{21} = A_{24} = A_{25} = A_{26} = 0$$

$$A_{22} = \ddot{\theta}_1 + \ddot{\theta}_2$$

$$A_{23} = L_1(\ddot{\theta}_1\cos(\theta_2) + \dot{\theta}_1^2\sin(\theta_2)) + g\sin(\theta_1 + \theta_2)$$

$$A_{27} = \text{sgn}(\dot{\theta}_2)$$

$$A_{28} = \dot{\theta}_2$$

Similarly, the specific form of the parameter vector to be identified can be expressed as:

$$X = \begin{bmatrix} X_1 & X_2 & X_3 & X_4 & K_{S1} & K_{m1} & K_{S2} & K_{m2} \end{bmatrix}^T$$

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
