# Peer review of "Modelling and RBF Control of Low-Limb Swinging Dynamics of a Human–Exoskeleton System"

_actuators, doi:10.3390/act12090353_

Round 1
Reviewer 1 Report (Previous Reviewer 2)
The authors made significant revisions, and the clarity of the contributions is now close to enough. However, the reviewer has some additional minor comments.
- There seem to be some grammatical errors. For example, the first line of the 3rd paragraph of Sec. 2.1 seems incomplete as a sentence. Also, the reviewer thinks that something is wrong with the sentence around Eqs. (9) and (10). Although the reviewer cannot be sure because he is not a native English speaker, he recommends checking the whole manuscript.
- The symbol T in the l.h.s. of Eq. (12) seems not defined.
- Fourth line of p. 10: The reviewer does not understand what they compared with the error in RBF control of the hip joint.
- From the discussion in Sec. 3.2, there seem to be some uncertainties introduced in their simulation. However, the reviewer does not understand what they are. They will be one of the most critical parts to demonstrate the effectiveness of the RBF control. Therefore, he believes that authors must clarify the origin of the uncertainties of the simulation.
Author Response
Please see the attachment.

Reviewer 2 Report (New Reviewer)
This work proposes a parameter identification strategy of a human lower-limb coupled with an exoskeleton and a control system based on a RBF neural network. In general, the work is interesting, but this reviewer have some concerns.
The style of the paper should be improved. In the text there are some intricate sentences and typos that hinder the readability of the manuscript. An extensive review of the English language is encouraged by this reviewer. Why do you use red text font in some sections of the manuscript?
This reviewer assumes that E_k and E_p (in line 108) are kinematic and potential energy, but this variables are not introduced in the text.
In section 2.1, a strong emphasis is made on the derivation of the dynamical model of the system. However this derivation do not provide relevant information to this work, since the derivation of the dynamical model of a RR kinematic chain has been extensively studied in the literature.
In Figure 1, distances L1, L2, c1, and c2 are not clearly indicated.
How were the values for the constraint intervals in Table 2 determined?
In equation (21), \Lambda is introduced but not referenced in the text.
In line 224 it is stated that "The chosen RBF neural network is a three-layer feedforward network with a single hidden layer". Why did you use this neural network?
In line 108 the Lagrange function was denoted by "L", and in equation (32) the Lyapunov function is denoted by "L" too.
At the beginning of section 3.1, the acronym "PSO" was used but not introduced.
How many subjects participated in the identification experiment? The subjects were sitting or standing? It is not clear how the identification experiment was performed. Please, consider the addition of a figure showing the experimental setup.
How is the exoskeleton used during the identification experiment? Has it been already published? Has it been developed for this work? Some information about the exoskeleton used during the experiments would improve the repeatability of this paper.
In section 4, a more extensive discussion of the results would improve the quality of the work. Besides, section 4 should be named "Discussion and Conclusions".
Some sentences might benefit from further restructuring to enhance readability and flow. Consider breaking down complex sentences and clarifying concepts as needed.
In some sentences, the same word appears several times. Consider the use of synonyms.
There are also some typos in the text.
Round 2
Reviewer 2 Report (New Reviewer)
The authors have address properly all my comments.
The readability of the manuscript has improve, but there still are some complex sentences.
This manuscript is a resubmission of an earlier submission. The following is a list of the peer review reports and author responses from that submission.
Round 1
Reviewer 1 Report
This paper studies the modelling, parameter identification, and the trajectory tracking control strategies of a wearable exoskeleton human-machine coupling system. The topic of this study is very interesting, and also of great prospects of application. The structure of this paper is also good and systematic. The reviewer hence recommends this paper to be published, with some suggestions as follows:
1) In the introduction, the renovation of this paper can be better introduced, like comparing the present work and existing works.
2) English can be polished. For example, line 34 and line 35 may be repetitive.
3) Format of notations. For example, a) line 104, N should be italic, same for T in line 114; b) the word "where" in line 195; c) line 105, the dimension of \bar{A} and \bar{\tau} can be added; d) In the control of RBF neural network, Kv is used, while Kv1 and Kv2 is used for friction coefficients of the system, perhaps different notation of Kv can be used.
4) Line 121, the Particle Swarm Optimization (PSO) algorithm is not cited.
See the above Comments and Suggestions for Authors.
Author Response
Actuators
Manuscript actuators-2506613: Control of low-limb swinging dynamics of a human-exoskeleton system by Xinyu Peng, Shujun Zhang, Mengling Cai and Yao Yan.
Response to the reviewers’ comments
The authors would like to thank all the reviewers and the editor for a careful reading of the manuscript and useful comments. We have revised the manuscript addressing the referee´s comments. A response to each of the comments is given below.
Comments:
Reviewer #1: This paper studies the modelling, parameter identification, and the trajectory tracking control strategies of a wearable exoskeleton human-machine coupling system. The topic of this study is very interesting, and also of great prospects of application. The structure of this paper is also good and systematic. The reviewer hence recommends this paper to be published, with some suggestions as follows:
1) In the introduction, the renovation of this paper can be better introduced, like comparing the present work and existing works.
Thank you so much for your kind suggestion. In this study, the inertial parameters of the human-machine coupling model are determined through identification experiments. The novelty of this work lies in not using anatomical parameters directly in the model simulation. This modification has been made at the end of the Introduction section. The introduction has been modified and some references has been added. Please find the revised manuscript for more details.
- English can be polished. For example, line 34 and line 35 may be repetitive.
Thank you so much for your kind suggestion. The English expression has been modified and the duplicated parts have been removed. Please find the revised manuscript for more details.
- Format of notations. For example, a) line 104, N should be italic, same for T in line 114; b) the word "where" in line 195; c) line 105, the dimension of \bar{A} and \bar{\tau} can be added; d) In the control of RBF neural network, Kv is used, while Kv1 and Kv2 is used for friction coefficients of the system, perhaps different notation of Kv can be used.
Thank you so much for your kind suggestion.
- a) Line 104 of N has been changed to italicized, and line 117 of T has also been changed to italicized. In addition, similar errors were checked and modifications were made throughout the entire text;
- b) The ‘where’ in line 195 has been modified;
- c) In line 105, the dimension of two parameters has been added;
- d) Kv1 and Kv2 have been modified to Km1 and Km2;
Please find the revised manuscript for more details.
- Line 121, the Particle Swarm Optimization (PSO) algorithm is not cited.
Thank you so much for your kind suggestion. The particle swarm algorithm is merely used as a method to optimize the trajectory, and the condition number of the sampled regression matrix is chosen as the fitness function. Therefore, the article does not focus on describing the particle swarm algorithm in detail. Minor modifications have been made in this section(after the table 2). Please find the revised manuscript for more details.

Reviewer 2 Report
This paper considers the control of a wearable lower-limb exoskeleton robot. It proposes a method to estimate physical parameters from experimental data and compare the performance of two control strategies.
The reviewer has two main concerns: one is that the contribution is unclear, and the other is that the comparison results seem insignificant.
Regarding the first point, the authors cite many related studies in Introduction, but the relation with this paper is unclear. Especially, there is no mention of what is novel compared with existing ones. A similar problem is also in the Abstract: there are no statements of the state-of-the-art, but only explain what they have done. Therefore, it is hard to understand why this study is necessary.
Regarding the second point, they showed that, for example, by enlarging the gains, they can reduce the tracking error, but increasing the gains will result in noise amplification. However, it is well known (written in most textbooks) that a higher P gain will result in a smaller residual because it shifts the gain plot upward, and a higher D gain may cause noise amplification because it enlarges the gain at the high-frequency region. Regarding the RBF controller tau=K_v*q-v with q defined as Eq. (10), the effect of K_v can be understood in the same way as in PD control because the first term represents a special instance of PD control. Furthermore, the reviewer suspects that the controller performance is different between the real machine and the simulation, which may make the comparison of the two controllers meaningless. Therefore, for the reviewer, the contents of the Result section seem not novel nor significant.
Detailed comments:
- The reviewer does not understand the meaning of c_1 and c_2, which seem not used. Are they the position or simply the names given to the center of the masses?
- Please be consistent with the use of italic fonts for mathematical symbols. Some are used both in italic and upright fonts.
- Eq. (6) seems weird: a table is not a set.
- In the system identification, it says that the PSO algorithm is used to optimize the excitation trajectory. However, the reviewer suspects they need one trajectory to assess the objective function at one set of parameters. As PSO requires many samples in general, it will require a massive amount of experiments. Therefore, the reviewer wonders how the authors obtained enough samples, or whether they actually obtained sufficient data. Note that PSO is good at solving high-dimensional optimization problems, but, in general, not that efficient for lower-dimensional ones like the one in this study (the objective function has 6-dimensional input).
- The reviewer finds the definition of \Lambda is ambiguous. Is it a design parameter, or is given by some theoretical deduction?
- Line 249 of p. 7: "but the rate of decrease becomes slower over time"
The reviewer was confused by this sentence because there is no mention of time evolution before this sentence.
- The caption of Figure 4: Please avoid inserting page-break before the caption.
Author Response
Actuators
Manuscript actuators-2506613: Control of low-limb swinging dynamics of a human-exoskeleton system by Xinyu Peng, Shujun Zhang, Mengling Cai and Yao Yan.
Response to the reviewers’ comments
The authors would like to thank all the reviewers and the editor for a careful reading of the manuscript and useful comments. We have revised the manuscript addressing the referee´s comments. A response to each of the comments is given below.
Comments:
Reviewer #2: This paper considers the control of a wearable lower-limb exoskeleton robot. It proposes a method to estimate physical parameters from experimental data and compare the performance of two control strategies.
The reviewer has two main concerns: one is that the contribution is unclear, and the other is that the comparison results seem insignificant.
Regarding the first point, the authors cite many related studies in Introduction, but the relation with this paper is unclear. Especially, there is no mention of what is novel compared with existing ones. A similar problem is also in the Abstract: there are no statements of the state-of-the-art, but only explain what they have done. Therefore, it is hard to understand why this study is necessary.
Regarding the second point, they showed that, for example, by enlarging the gains, they can reduce the tracking error, but increasing the gains will result in noise amplification. However, it is well known (written in most textbooks) that a higher P gain will result in a smaller residual because it shifts the gain plot upward, and a higher D gain may cause noise amplification because it enlarges the gain at the high-frequency region. Regarding the RBF controller tau=K_v*q-v with q defined as Eq. (10), the effect of K_v can be understood in the same way as in PD control because the first term represents a special instance of PD control. Furthermore, the reviewer suspects that the controller performance is different between the real machine and the simulation, which may make the comparison of the two controllers meaningless. Therefore, for the reviewer, the contents of the Result section seem not novel nor significant.
Thank you so much for your kind suggestion. This work adopts a model-based control approach, which requires an accurate model. Therefore, this study first uses identification methods to determine the model parameters instead of using average anatomical values for human body parts. This is the innovation that sets this research apart from others. Additionally, the reference list includes descriptions of other teams' work using RBF for robot control, which makes the use of RBF adaptive control relevant for subsequent experimental research. In the following research, we will compare the two control schemes based on simulation results on an experimental platform for real experiments. The performance of the controller will be verified through real experimental data.
1) The reviewer does not understand the meaning of c_1 and c_2, which seem not used. Are they the position or simply the names given to the center of the masses?
Thank you so much for your kind suggestion. c1 and c2 represent the thigh center of mass and shank center of mass, respectively, I give the description in Table 1.
- Please be consistent with the use of italic fonts for mathematical symbols. Some are used both in italic and upright fonts.
Thank you so much for your kind suggestion. The italics in the full text Mathematical symbols have been revised and checked. In addition, similar errors were checked and modifications were made throughout the entire text. Please find the revised manuscript for more details.
- (6) seems weird: a table is not a set.
Thank you so much for your kind suggestion. Equation (6) has been modified, then it is a equation. Please find the revised manuscript for more details.
- Line 121, the Particle Swarm Optimization (PSO) algorithm is not cited. In the system identification, it says that the PSO algorithm is used to optimize the excitation trajectory. However, the reviewer suspects they need one trajectory to assess the objective function at one set of parameters. As PSO requires many samples in general, it will require a massive amount of experiments. Therefore, the reviewer wonders how the authors obtained enough samples, or whether they actually obtained sufficient data. Note that PSO is good at solving high-dimensional optimization problems, but, in general, not that efficient for lower-dimensional ones like the one in this study (the objective function has 6-dimensional input).
Thank you so much for your kind suggestion. The use of PSO algorithm for trajectory optimization involves sampling the trajectory to be optimized (equation 5) in MATLAB. Although the efficiency of PSO Algorithmic efficiency is not high, the use of optimized trajectory in subsequent experiments can get better data from the experimental results for the next step. In the ‘Result’ section of the identification experiment, a simulation sampling explanation of the PSO algorithm was added. Please find the revised manuscript for more details.
5、The reviewer finds the definition of \Lambda is ambiguous. Is it a design parameter, or is given by some theoretical deduction?
Thank you so much for your kind suggestion. It is a design parameter obtained from reference [27].
6、Line 249 of p. 7: "but the rate of decrease becomes slower over time"
The reviewer was confused by this sentence because there is no mention of time evolution before this sentence.
Thank you so much for your kind suggestion. This sentence has been revised to read as “but as the gain increases”. Please find the revised manuscript for more details.
7、 The caption of Figure 4: Please avoid inserting page-break before the caption
Thank you so much for your kind suggestion. The page-break has been removed. Please find the revised manuscript for more details.

Reviewer 3 Report
The author provides a detailed description of the theoretical and experimental steps of the identification experiment, and identifies the necessary parameters for PD control and RBF control simulation through the experiment. Afterwards, the effects of PD control and RBF control were compared, and detailed pictures were provided for demonstration. Finally, it was concluded that RBF control had better effects than PD control.
Generally, the author’s idea is described clearly, but some revisions should be made before the publication. Relevant questions and suggestions are listed as follows:
1. The identification experiment involves the human body, and there is no ethical statement in the paper. Whether ethical review approval has been obtained, please provide an ethical statement in the appropriate position of the article;
2. Please use general text to represent "where" in formulas (10), (17), and (19), and do not write it in MathType;
3. Pay attention to the layout of Table 3. Do not divide a table into two pages. If you want to divide it into two pages, please indicate Continuation 3 on the table on the second page;
4. What are the units of Kp and Kd in Figure 2? Please indicate them in the figure;
5. Each subgraph in Figure 3 has only one line, and the "RBF" in the upper right corner can be removed;
6. Pay attention to the layout of Figure 4 and do not separate the Figure and title on two pages;
7. The "(N · m)" in the upper right corner of Figure 6 that explains the line can be removed;
8. In the appendix A, "L1=0.38952" should be indicated in italics, corresponding to the formula and the main text.
The English writing needs to be well optimized.
Author Response
Actuators
Manuscript actuators-2506613: Control of low-limb swinging dynamics of a human-exoskeleton system by Xinyu Peng, Shujun Zhang, Mengling Cai and Yao Yan.
Response to the reviewers’ comments
The authors would like to thank all the reviewers and the editor for a careful reading of the manuscript and useful comments. We have revised the manuscript addressing the referee´s comments. A response to each of the comments is given below.
Comments:
Reviewer #3: The author provides a detailed description of the theoretical and experimental steps of the identification experiment, and identifies the necessary parameters for PD control and RBF control simulation through the experiment. Afterwards, the effects of PD control and RBF control were compared, and detailed pictures were provided for demonstration. Finally, it was concluded that RBF control had better effects than PD control.
Generally, the author’s idea is described clearly, but some revisions should be made before the publication. Relevant questions and suggestions are listed as follows:
1) The identification experiment involves the human body, and there is no ethical statement in the paper. Whether ethical review approval has been obtained, please provide an ethical statement in the appropriate position of the article;
Thank you so much for your kind suggestion. An ethical statement was made before Table 3 in the article. Please find the revised manuscript for more details.
2)Please use general text to represent "where" in formulas (10), (17), and (19), and do not write it in MathType;
Thank you so much for your kind suggestion. The 'where' of formulas (10), (17), and (19) has been changed to general text. Please find the revised manuscript for more details.
3)Pay attention to the layout of Table 3. Do not divide a table into two pages. If you want to divide it into two pages, please indicate Continuation 3 on the table on the second page;
Thank you so much for your kind suggestion. Table 3 has been placed on the same page. Please find the revised manuscript for more details.
4) What are the units of Kp and Kd in Figure 2? Please indicate them in the figure;
Thank you so much for your kind suggestion. Units have been added to Kp and Kd in Figure 2. Please find the revised manuscript for more details.
5) Each subgraph in Figure 3 has only one line, and the "RBF" in the upper right corner can be removed;
Thank you so much for your kind suggestion. The 'RBF' in the upper right corner has been removed. Please find the revised manuscript for more details.
6) Pay attention to the layout of Figure 4 and do not separate the Figure and title on two pages;
Thank you so much for your kind suggestion. The figure and title of Figure 4 have been placed on the same page. Please find the revised manuscript for more details.
7) The "(N · m)" in the upper right corner of Figure 6 that explains the line can be removed;
Thank you so much for your kind suggestion. The ‘(N · m)’ in the upper right corner of Figure 6 that explains the line have been removed. Please find the revised manuscript for more details.
8) In the appendix A, "L1=0.38952" should be indicated in italics, corresponding to the formula and the main text.
Thank you so much for your kind suggestion. L1 has been represented in italics. Please find the revised manuscript for more details.

Reviewer 4 Report
The article “Control of low-limb swinging dynamics of a human-exoskeleton system” addresses the important topic of lower limb exoskeleton control. However, there are several areas that require potential improvement and consideration.
Point 1. The introduction provides clear information about the aging population and the need for lower limb exoskeletons. However, it is missing detailed information about existing modern exoskeleton controls, their limitations, and why the approach presented in this study is preferable. Including a comprehensive review of existing exoskeleton control strategies would enhance the context and rationale for the proposed approach.
Point 2. In line 148, a formula is presented without an explanation of the variables, such as Kp and Kd. While the explanation is provided later in the text, it can be confusing for readers. It would be beneficial to include a reference or a brief explanation of these variables when first introduced to aid understanding.
Point 3. The article includes a graphical comparison of the PD and RBF control methods. However, there is a lack of numerical or quantitative results presented alongside the visual representations. Additionally, it would be valuable to discuss whether the observed improvement is sufficient compared to other state-of-the-art control methods.
Point 4. The discussion section does not effectively serve as a discussion. It lacks a thorough exploration of the results, limitations, comparisons with other works, and suggestions for future research. he absence of references to other related studies. Expanding the discussion to include these elements would improve the article.
Author Response
Actuators
Manuscript actuators-2506613: Control of low-limb swinging dynamics of a human-exoskeleton system by Xinyu Peng, Shujun Zhang, Mengling Cai and Yao Yan.
Response to the reviewers’ comments
The authors would like to thank all the reviewers and the editor for a careful reading of the manuscript and useful comments. We have revised the manuscript addressing the referee´s comments. A response to each of the comments is given below.
Comments:
Reviewer #4: The article “Control of low-limb swinging dynamics of a human-exoskeleton system” addresses the important topic of lower limb exoskeleton control. However, there are several areas that require potential improvement and consideration.
1) The introduction provides clear information about the aging population and the need for lower limb exoskeletons. However, it is missing detailed information about existing modern exoskeleton controls, their limitations, and why the approach presented in this study is preferable. Including a comprehensive review of existing exoskeleton control strategies would enhance the context and rationale for the proposed approach.
Thank you so much for your kind suggestion. The introduction section has been modified and some references have been added to provide a detailed explanation of the control section. Please find the revised manuscript for more details.
2) In line 148, a formula is presented without an explanation of the variables, such as Kp and Kd. While the explanation is provided later in the text, it can be confusing for readers. It would be beneficial to include a reference or a brief explanation of these variables when first introduced to aid understanding.
Thank you so much for your kind suggestion. KP and Kd have been explained after the formula. Please find the revised manuscript for more details.
3) The article includes a graphical comparison of the PD and RBF control methods. However, there is a lack of numerical or quantitative results presented alongside the visual representations. Additionally, it would be valuable to discuss whether the observed improvement is sufficient compared to other state-of-the-art control methods.
Thank you so much for your kind suggestion. You are definitely right. It is a great idea to address the significance of this study. The simulation results comparing PD control and RBF control have been modified to include both graphical comparisons and numerical values for quantitative analysis. The introduction section cites references that explain the use of RBF for controlling lower limb exoskeletons, highlighting the significance of employing RBF for controlling such exoskeletons. This work focuses solely on control simulations under passive control without considering the wearer's intent. In future work, the wearer's intent will be taken into account, and improvements will be made to the control algorithm.
4) The discussion section does not effectively serve as a discussion. It lacks a thorough exploration of the results, limitations, comparisons with other works, and suggestions for future research. he absence of references to other related studies. Expanding the discussion to include these elements would improve the article.
Thank you so much for your kind suggestion. In this study, we assumed that the patient does not exert any force and is completely driven by the exoskeleton, without considering the patient's intent. In future research, we will take the patient's movement intent into account and conduct in-depth investigations into the influencing factors. We plan to carry out further research on active control in the future. Please find the revised manuscript for more details.

Round 2
Reviewer 2 Report
The authors made significant revisions but they seem to address neither of the reviewer's main concerns in an adequate manner.
- The authors wrote in their response that ``This is the innovation that sets this research apart from others'' regarding their system identification. However, the reviewer does not find supporting evidence in the manuscript. As system identification has been a hot topic in some communities including control theorests for some decades, he believes that they have to address those studies and describe what is innovating in the manuscript.
- If, as the authors claim in their response, the system identification is one of the main contributions, the reviewer believes that the experiments must illustrate its validity/efficacy. This will require some additional experiments.
- The reviewer still does not understand the necessity of the comparison of the controllers described in the manuscript. He understands the potential importance of RBF adaptive control, but why is it necessary to perform another comparison with the classic PID controller? Furthermore, as the RBF controller seems to be adopted to deal with uncertainties, he does not believe simple simulations can illustrate its effectiveness.